# Gene Content and Coding Diversity of the Growth Hormone *Loci* of Apes

**DOI:** 10.3390/genes14020241

**Published:** 2023-01-17

**Authors:** Rafael González-Álvarez, Irám Pablo Rodríguez-Sánchez, Hugo A. Barrera-Saldaña

**Affiliations:** 1Instituto Tecnológico de los Altos de Jalisco, Yahualica de González Gallo, Zapopan 47300, Mexico; 2Facultad de Ciencias Biológicas, Laboratorio de Fisiología Molecular y Estructural, Universidad Autónoma de Nuevo León, San Nicolás de los Garza 66455, Mexico; 3Vitagénesis, SA de CV/Innbiogem, Monterrey 64630, Mexico; 4Facultades de Medicina y Ciencias Biológicas, Universidad Autónoma de Nuevo León, San Nicolás de los Garza 66455, Mexico

**Keywords:** *Chorionic somatomammotropins*, *Somatolactins*, *Prolactin*, *Platyrrhine*, *Cercopithecidae*, monkeys

## Abstract

The growth hormone (GH) *locus* has experienced a dramatic evolution in primates, becoming multigenic and diverse in anthropoids. Despite sequence information from a vast number of primate species, it has remained unclear how the multigene family was favored. We compared the structure and composition of apes’ *GH loci* as a prerequisite to understanding their origin and possible evolutionary role. These thorough analyses of the *GH loci* of the chimpanzee, gorilla, and orangutan were done by resorting to previously sequenced bacterial artificial chromosomes (BACs) harboring them, as well as to their respective genome projects data available in GenBank. The GH *loci* of modern man, Neanderthal, gibbon, and wild boar were retrieved from GenBank. Coding regions, regulatory elements, and repetitive sequences were identified and compared among species. The *GH loci* of all the analyzed species are flanked by the genes *CD79B* (5′) and *ICAM-1* (3′). In man, Neanderthal, and chimpanzee, the *loci* were integrated by five almost indistinguishable genes; however, in the former two, they rendered three different hormones, and in the latter, four different proteins were derived. Gorilla exhibited six genes, gibbon seven, and orangutan four. The sequences of the proximal promoters, enhancers, P-elements, and a *locus* control region (LCR) were highly conserved. The locus evolution might have implicated duplications of the ancestral pituitary gene (*GH-N*) and subsequent diversification of the copies, leading to the placental single *GH-V* gene and the multiple *CSH* genes.

## 1. Introduction

Growth hormone (GH), placental lactogen (PL), prolactin (PRL), prolactin 2 (PRL2), and somatolactin (SL) in vertebrates constitute a family of proteins within the large superfamily of class-1 helical cytokines [1]. They all display a characteristic conserved amino-acid framework with four cysteine residues forming two disulfide bridges that stabilize their four-helical tertiary structure [2,3,4]. Even though there is considerable variation among members of this family in the evolutionary rate across human and non-human primates, and distant members share only about 25–30% amino acid sequence identity, their conserved cysteine framework, the exon organization of their genes, and the binding properties of the members to their respective receptors define them as a monophyletic family within the cytokine superfamily [1]. In primates, the *GH locus* experienced a dramatic evolution, resulting in a gene cluster coding at least three types of hormones: the pituitary one (GH-N), its placental variant (GH-V), and the chorionic somatomammotropin hormone (CSH, previously referred to as placental lactogen), which appeared for the first time in *Cercopithecidae* (old world monkeys and apes, including humans). Despite the availability of considerable gene sequence information from a vast number of primate species, it has remained unclear how the gene family formed, and different proposals have been discussed extensively over the years [5]. A predominant hypothesis states that gene duplications of an ancestral version of the *PRL* gene gave rise to two branches, the *PRL* itself and the *GH* [6]. In mammals, except primates, the *PRL* branch gave rise to placental lactogens, and in the latter (primates), *CSHs* emerged from the *GH* branch [7].

While most mammalian species have a single gene that encodes GH, in the primate superfamilies *Platyrrhine* (new world monkeys) and *Cercopithecidae* (old world monkeys), gene clusters emerged and diverged independently via dramatic events of gene duplication and conversion [8,9,10]. In fact, at the bottom of the primate evolutionary tree in prosimians, such as the lemur (*Lemur catta*), the GH proteins show a higher sequence similarity between them and the GH of non-primate mammals, such as the wild boar (*Sus scrofa*), than with the GHs of higher primates. Thus, it is inferred that the accelerated series of evolutionary events that characterize this gene family began after separating the prosimian and higher primates’ lineages (see Figure 1). This finding could be due to positive selection on the original gene, related to adaptive changes in the functioning of its encoded hormone, especially in its metabolic support of reproduction [11,12].

These events of gene duplication, gene conversion, and the diversification of the new genes derived from the ancestral *GH* gene in primates constitute a dramatic evolutionary story that makes this locus an extraordinary case for molecular evolution studies. Their implications span from the emergence of spatial and temporal control of gene expression to a still non-resolved role of the multiple new encoded hormones in the endocrinology of pregnancy and its possible implications on the survival of the human fetus [14].

GH performs its biological activity among species via the actions of the somatotrophic axis [15]. Human disorders, including reduced stature and delayed sexual maturity, can result when the normal actions of HGH are disrupted [16,17]. Similar to most higher anthropoids, humans are characterized by prolonged gestation and delayed fetal maturation rates, with many anthropoid newborns having large brains relative to their body sizes. These features have been advanced as the basis for increased social complexity and cognitive capacity in primates [15]. The genetic basis of these characteristic anthropoid phenotypes is unknown; however, fetal development depends on access to maternal resources during pregnancy. Indeed, it has recently been shown that hemochorial placentation in anthropoids is associated with steeper brain-body allometry, faster prenatal brain growth, and slower prenatal body growth [18]. The placental variant of GH (HGH-V) and CSHs have been implicated in the fetal acquisition of maternal resources during anthropoid pregnancy [19]. *GH*-related genes that have emerged in primates, such as those in man (*CSH-L*, *CSH-A*, *GH-V*, and *CSH-B*) [8] and rhesus macaque (*GH-2, CSH-1, CSH-2, CSH-3,* and *CSH-4*) [20], and most likely in the rest of groups of primates, are transcribed in the placenta. Although largely uncharacterized, these hormones derived from these placenta-expressed genes have been implicated in diverse roles during pregnancy, from mediating trophoblast invasion [21] to regulating maternal resource availability for the developing fetus [22].

It should be noted that the expression of *GH locus* elements is not limited to the pituitary gland or the placenta. Since the last quarter of the past century, studies have demonstrated the presence of GH in several extra-pituitary sites, such as neural, ocular, reproductive, immune, cardiovascular, muscular, dermal, and skeletal tissues [23]. For instance, the presence of GH in ocular structures, such as the retina, choroid, and cornea, seems to function as an emergency mechanism against injuries. Because of this, GH can be used as a biomarker to follow retinal neurodegeneration [24]. This finding is an intriguing case that reflects the need to investigate new physiological functions of GH and to unravel the surely complex mechanisms that regulate its extra-pituitary gene expression.

Thus, studying the evolutionary history of these gene clusters uniquely shared among anthropoids can illuminate important aspects of human pregnancy, fetal development, and other unknown physiological processes. Given that these physiological features are of recent acquisition in evolution, the putative contribution of the GH family must be reflected in its evolutionary history in primates. Therefore, to evaluate the evolution of GH in primates, we examined and compared the composition and structure of the *GH loci* in apes’ species.

## 2. Materials and Methods

### 2.1. Sequences

*GH loci* sequences from the human (*Homo sapiens*), Neanderthal (*Homo sapiens neanderthalensis*), chimpanzee (*Pan troglodytes*), gorilla (*Gorilla gorilla*), orangutan (*Pongo abelli*), gibbon (*Nomascus leucogenys*), and wild boar (*S. scrofa*) were retrieved from GenBank (Table 1). We also relied on the chimpanzee, gorilla, and orangutan sequences obtained before from the BACs containing the *GH locus*. These BACs were sequenced by NGS using the Roche 454 platform, as previously described [25,26].

### 2.2. Genomic Annotations

The annotation of genetic elements (such as genes and their exons and introns) of the nucleotide sequences analyzed in this work was carried out using SnapGene 5.2 software. Repetitive elements, predominantly present in intergenic regions of the *GH locus*, were identified with the RepeatMasker 4.0.9 program [27] with the default parameters (HMMER o NCBI/RMBLAST search methods). Regulatory elements were searched by similarity with the previously characterized ones [25,26], identifying the Sp1 binding site, the Pit-1 distal and proximal binding sites, the thyroid hormone response element (TRE), the initiator element (InrE), the cyclic AMP response element (CRE), and the TATA box, mainly.

### 2.3. Phylogenetic Analysis

The sequences were aligned using the Clustal Omega program [28], followed by manual corrections if needed. Protein sequences were derived by conceptual translation of the coding sequences. A phylogenetic tree was built from coding sequences with MEGA 6.06 software [29] using the Neighbor-Joining (NJ) method. Then, a bootstrap test was done with 1000 replicates [30]. The GenBank accession numbers for the *GH loci* (together with their flanking genes) used in this study are listed in Table 1. Wild boar (*S. scrofa*) was used as the external group for the evolutionary analyses.

## 3. Results

### 3.1. Organization of the Primate GH loci

The *GH locus* in higher primates invariably contains the *GH-N* gene, the *GH-V* gene, and a variable number of *CSH* genes. The *loci* of all the species investigated are flanked by the genes *CD79B* and *ICAM-1*, including the one of the wild boar, as shown in Figure 2. The *locus* contains five genes in man, Neanderthal, and chimpanzee, while it harbors six genes in the gorilla, four in the orangutan, and seven in the gibbon; as expected, the wild boar carries a single *GH-N* gene. It is noteworthy that the *GH-N*, *GH-V*, and *CSH-B* genes are invariably present in all of the higher primates and that the differences between their *loci* lie in the diversity of the *CSH-A* gene. In modern man, Neanderthal, orangutan, and gibbon, the gene immediately following *GH-N*, the *CSH-L,* carries a splicing mutation, becoming a pseudogene (see Figure 2).

The *GH loci* have between 63–98 Kb, 15–29 *Alu* sense, and 15–28 *Alu* antisense in the five compared samples. Retrotransposons are present in orangutan (one) and chimpanzee (two) (Table 2).

### 3.2. Phylogenetic Analyses

A phylogenetic tree was constructed based on the cDNA sequences of the *GH*/*CSH* genes using the Neighbor-Joining (NJ) method, relying on one sequence as an outgroup (wild boar). As depicted in Figure 3 its branches show well-defined clades, such as *GH-N*, *GH-V*, *CSH-A/B*, and the pseudogene *CSH-L*.

Interestingly, the gibbon *GH locus* shows two *GH-N* genes (Figure 2). They make a clade with the *GH-N* genes (Figure 3), suggesting that this deviation of the patterns of gene composition for the *loci* of all the species of primates analyzed so far may be due to an annotation mistake, as this genome is still a draft. The presence of two *GH-N* or *GH-V* genes is very improbable.

### 3.3. Regulatory Elements

As described previously [25,26], besides repetitive and transposable elements, regulatory elements in the promoters, enhancers, and a *locus* control region (LCR, outside the *GH* locus limits) were identified by sequence similarity (Figure 4). The promoter regions are conserved in the different genes in all of the species. The Sp1 element and the TATA box are completely conserved, while the distal Pit-1 is highly conserved, identical in the *GH-N* and the *GH-V* genes, but showing only one nucleotide substitution in the *CSH* genes. On the other hand, substitutions are observed in the proximal Pit-1 binding site in the *GH-V* genes. Likewise, the TRE shows considerable variation among the genes. On the other hand, CRE is well conserved, having only two substitutions in the orangutan *GH-N* gene, one in the gorilla *CSH-B* gene, and one in each of the orangutan *GH-V*, *CSH-L*, and *CSH-B* genes. The InrE shows substitutions mainly in the *CSH* and the *GH-V* genes (Appendix A).

The placental enhancers are located downstream of the *CSH* genes. These are well-conserved and display all the typical organizations in four domains (DF-1 to DF-4). There are only a few substitutions in its DF-1 and DF-2 domains. A substitution is present in DF-1 (position 16) in the *CSH-B* genes’ enhancers (except for the orangutan). These enhancers also show deletions at the upstream region of the DF-2 domain in man, Neanderthal, chimpanzee, and gorilla. Moreover, a substitution in position 65 is also present in these enhancers. The DF-3 and DF-4 domains show a few mutations, particularly a substitution in DF-3 in position 228. Other mutations are also present in the primate *CSH* enhancer sequences, such as those in positions 112 and 134. However, these mutations are not inside these enhancer domains (Appendix A). The inhibitory P-element was also found to be highly conserved in all of the studied species. This element avoids *hCSH-L*, *hCSH-A*, *hGH-V*, and *hCSH-B* expression in the human pituitary gland. Thus, we can infer that the expression of the non-*GH-N* genes of the *GH loci* in the rest of non-human apes is excluded from their pituitary glands.

### 3.4. Gene Content and Diversity of Encoded Proteins

The apes’ *GH loci* genes code for three distinctive types of proteins: the pituitary GH-N, the placental GH-V, and CSHs. The *GH-N* and *GH-V* genes are constants, code for well-differentiated proteins, and each *locus* has only one copy. The protein encoded by the *GH-N* gene is the most well-conserved of all proteins coded by the apes’ *loci*, being identical in the human, Neanderthal, chimpanzee, gorilla, and orangutan. However, surprisingly, it has four to five changes in gibbon (see blue boxes in Appendix A). The protein encoded by the *GH-V* gene is also well-conserved, showing 0, 1, 4, 11, and 14 amino acid changes in the Neanderthal, chimpanzee, gorilla, orangutan, and gibbon, respectively (see pink boxes in Appendix A).

In contrast with the *GH* genes, the *CSH* genes differ in number and capacity to render protein diversity as the encoded proteins in a few of these genes are, in some cases, identical (such as in the case of the two functional genes of the human *locus*). As in the case of man, where the second gene (*hCSH-L*) harbors a splice donor mutation converting it into a pseudogene [31], its orthologous genes in the Neanderthal and orangutan also possess the same mutation and thus are inferred to be pseudogenes too. In addition, from the point of view of diversity, the proteins encoded by the *CSH* genes are more diverse, as shown in the orange boxes in Appendix A. While the *CSH* genes in number vary from one to four, the number of different encoded proteins does not follow the gene count. For example, in humans, there are *CSH* genes: *CSH-L*, *CSH-A*, and *CSH-B*, with the first being a pseudogene and the two remaining encoding the same mature protein (see red box in Appendix A). The gene count of three is reduced to a protein count of one. Thus, five *GH-CSH* human genes produce three proteins (one HGH-N, one HGH-V, and one CSH). A similar pattern is seen in the Neanderthal. In the chimpanzee, the *CSH-A1* gene produces the same mature protein as *CSH-A2* (red box in Appendix A). Its *GH locus* renders four proteins: one GH-N, one GH-V, and two different CSHs. Then, three different mature proteins are derived from the gorilla *CSH* genes as its *CSH-A2* gene produces the same protein as its *CSH-A3* gene (red box in Appendix A). Thus, the gorilla *GH locus* produces four proteins: one GH-N, one GH-V, and two CSHs. The orangutan *GH locus* has two *CSH* genes, but one is a pseudogene. Thus, the orangutan *GH locus* produces three proteins: one GH-N, one GH-V, and one CSH. Finally, the gibbon has four *CSH* genes; one is a pseudogene, and the rest do not show obvious mutations that could suggest a failure in their expression. Therefore, the gibbon *GH locus* apparently produces six proteins: two GH-N, one GH-V, and three CSHs. A summary of these gene counts versus the potentiality to render diverse proteins is depicted in Appendix A and Figure 5.

## 4. Discussion

Understanding the evolutionary forces that have shaped the multigene family of *GH* and *PRL* in mammals has been a subject of great interest. In this study, we investigated the gene composition and arrangement of anthropoideans’ *GH loci*. We inferred gene duplications and specialization events that could have given rise to the *GH* family in higher primates to explain their present-day anatomy. Our analysis is broader, covering the architecture of the *GH loci*, the comparison of key regulatory and repetitive elements, and the phylogenetic analyses of the genes.

The anatomy of the *GH loci* in the investigated ape species, as depicted in Figure 2, shows the organization of their gene members in the following general order: *GH-N*, *CSH-A* (variable number), *GH-V*, and *CSH-B*. The *loci* always begin with *GH-N*, followed by *CSH*s (including the *CSH-L*), *GH-V*, and finally, one more *CSH* gene. All of these *loci*, including all of the known non-primate *loci*, are flanked by the *CD79B* gene at the 5′ end and by the *ICAM-1* gene at the 3′ end. The *GH loci* of the analyzed ape species have several repetitive sequences of the *Alu* family in an average of one every 1000 bp. It is well known that this repetitive sequence favors the non-homologous recombination events that originate in multigene families. Given that, in primates, the flanking genes do not change much, *GH-N* always is the first gene, and because of the presence of numerous *Alu* sequences, it can be inferred that the duplications and nucleotide changes that originated in this family occurred only within the *GH loci*. How this happened might well have resulted from the alternative evolutionary routes proposed in Figure 6.

The phylogenetic analysis shown in the tree in Figure 3 displays four well-defined and well-supported clades by bootstrap analysis: from top to bottom, *GH-N*, *GH-V*, *CSH*, and non-primate *GH* (out-group). This indicates that *GH-N* is the orthologous gene, while *GH-V* and the *CSH*s are paralogous. This analysis supports the evolutionary pathway depicted in Figure 6, with two possible routes. In the first, the ancestral *GH-N* gene was duplicated to give rise to *GH-N*-like genes. Then, these experienced a series of mutations that gave rise to the placental genes (*GH-V* and *CSH*s) and later to the pseudogenization of a *CSH* gene. In a variation of this route, the ancestral *GH-N* was duplicated in one pre-*CSH* gene; then, in the second round of duplication, the block of *GH-N* and pre-*CSH* were duplicated into pre-*GH-V* and other pre-*CSH*. Likewise, further mutations in the pre-*GH-V* resulted in *GH-V*. Finally, the third duplication gave rise to two *CHS* genes, and one of them acquired a splicing mutation that converted the *CSH-L* gene into a pseudogene. 

Another striking feature of the analyzed *loci* is the diversity of their encoded hormones. We found that the *CSH* gene quantity did not match the CSH mature protein diversity they encoded. The human and Neanderthal have three *CSH* genes that encode only one CSH mature protein. The chimpanzee has three *CSH* genes that encode two CSH mature proteins, the gorilla has four *CSH* genes that encode three CSH mature proteins, and the orangutan has two *CSH* genes that encode one CSH mature protein. Finally, the gibbon has four *CSH* genes that encode three CSH mature proteins (Appendix A and Figure 5). This fact opens two interesting questions, such as (1) “Why do two genes encode the same protein?” and (2) “Do they produce the same protein quantities?”. The first question could be answered by the previously performed phylogenetic analyses [9,10,11,12,13,20,23,25,26,32]. They indicate that the evolutionary forces underlying the *GH locus* in higher primates are the purifying selection (dN < dS) typical of functional genes. Although we lack information to solve the second question in all apes, we explored it in humans and found that they produced similar amounts [33]. Although this could be an answer to the need to produce surprisingly high amounts of their encoded identical encoded proteins, which reach a couple of grams at the end of pregnancy [34], to answer these and other related questions fully, expression profiles in the apes’ hypophysis and placenta might be needed.

## 5. Conclusions

In this work, we determined the gene composition, organization, and proteins encoded in the *GH loci* of apes. The consensus arrangement of their genes is *GH-N*, *CSH-A* (variable number), *GH-V*, and *CSH-B*. The gene and protein contents are always a one-to-one proportion for the *GH-N* and *GH-V* type genes, while three-to-two or two-to-one for the *CSH* type genes. Finally, we proposed two hypotheses to explain the origin of these multigenic *loci*. First, a duplication event followed by events of mutation/specialization. Second is two duplications, followed by mutations/specializations and an additional duplication event. The most probable events that gave rise to the ape *GH loci* are described in scenario B. This scenario is the most plausible one as it explains better the pattern of the two pairs of *GH* and *CSH* genes that all *loci* harbor (see Figure 2).

## Figures and Tables

**Figure 1 genes-14-00241-f001:**
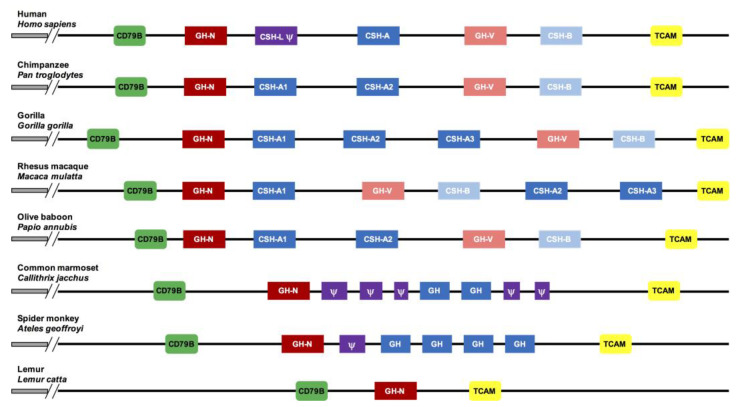
Dramatic evolution of the *GH locus* in primates. At the bottom is the lemur’s (a prosimian) single gene *locus* (*GH*, blue box), then, into new world monkeys (illustrated by the spider monkey and the marmoset), old world monkeys (exemplified by the baboon and the macaque), and apes (here demonstrated with the gorilla, chimpanzee, and human). The *locus* dramatically expands with the emergence of pseudogenes (Ψ), *GH-V*, and *CSH* genes. Adapted with permission from Refs. [11,12,13].

**Figure 2 genes-14-00241-f002:**
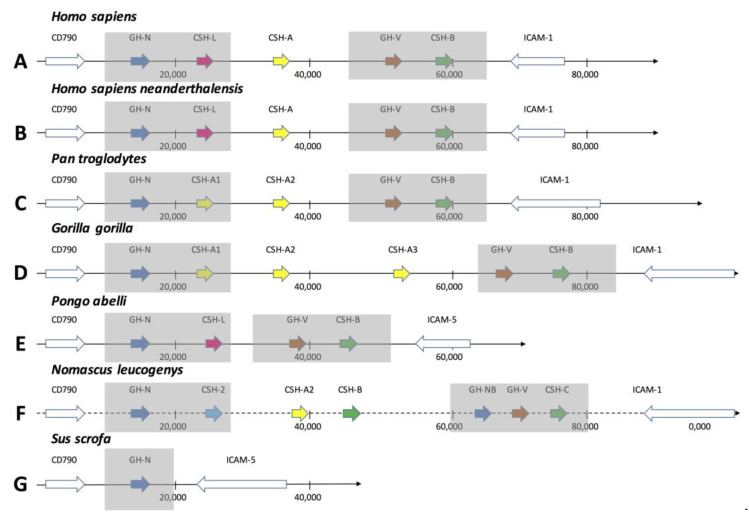
The apes’ *GH loci*. The genomic organization of the GH *loci* of man (*H. sapiens*, **A**), Neanderthal (*H. sapiens neanderthalensis*, **B**), chimpanzee (*P. troglodytes*, **C**), gorilla (*G. gorilla*, **D**), orangutan (*P. abelii*, **E**), gibbon (*N. leucogenys*, **F**), and wild boar (*S. scrofa*, **G**, used as external group for evolutionary analyses) is schematized. Blue, pink, yellow, orange, and green arrows indicate the *GH-N*, *CSH-L*, *CSH-A*, *GH-V*, and *CSH-B* genes, respectively. The white arrows indicate the genes *CD79B* and *ICAM-1* immediately flanking the *GH locus* in all of the assembly sequences reported in the present work. The direction of the arrows indicates the direction of the genes’ transcription. Continuous lines indicate complete sequences, while discontinuous ones represent missing sequences. Gray areas show the two pairs genes (except for *Sus scrofa*) that explain the most plausible evolutionary emergence scenario described in the Conclusions section.

**Figure 3 genes-14-00241-f003:**
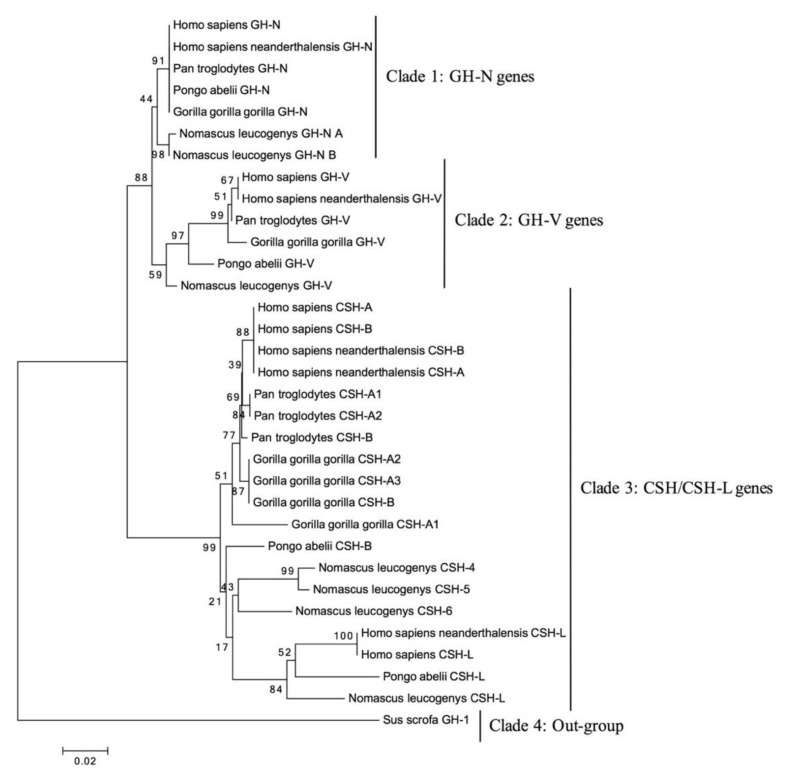
Phylogenetic tree. Neighbor-joining relationships were made in MEGA software based on the cDNA sequence of the *GH*/*CSH* genes of the human, Neanderthal, chimpanzee, gorilla, orangutan, and gibbon. For the outgroup, the sequence of wild boar (*S. scrofa*) was used to locate the tree root. Bootstrap percentages by 1000 replications are shown on the branches.

**Figure 4 genes-14-00241-f004:**
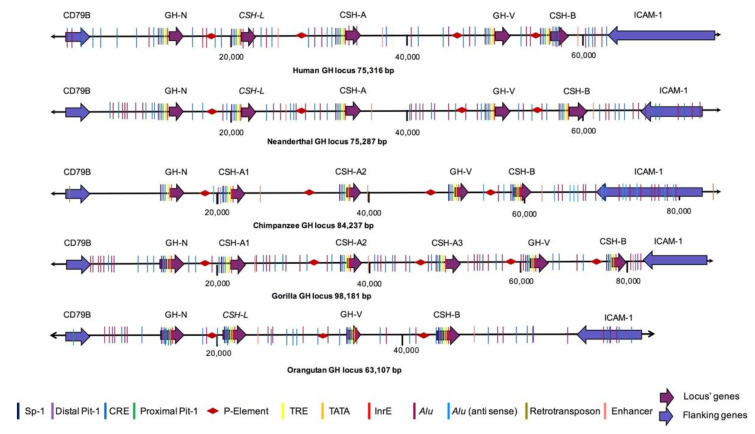
Hominid *GH loci*. The genomic organization of the *GH loci* for man (*H. sapiens*), Neanderthal (*H. sapiens neanderthalensis*), chimpanzee (*P. troglodytes*), gorilla (*G. gorilla*), and orangutan (*P. abelii*) is shown. The blue arrows indicate the *CD79B* and *ICAM-1* genes, while the purple ones represent the *GH* genes. The direction of the arrow indicates the direction of the genes’ transcription. Colored lines indicate the regulatory and repetitive elements.

**Figure 5 genes-14-00241-f005:**
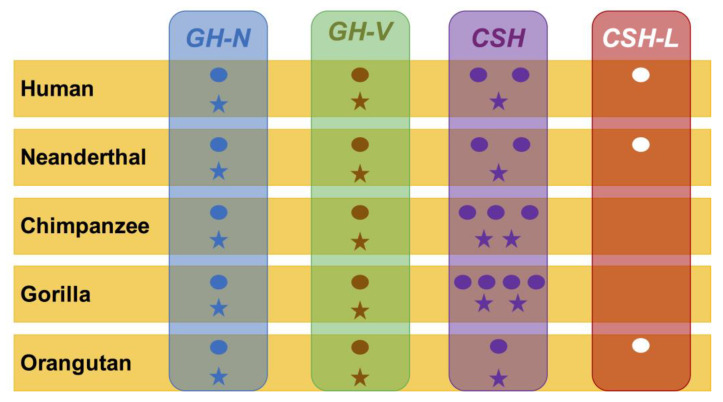
Gene content and diversity of hominid genes’ encoded proteins. For each species, the genes and proteins numbers are represented. The dots represent the gene content, and the stars indicate the number of encoded proteins.

**Figure 6 genes-14-00241-f006:**
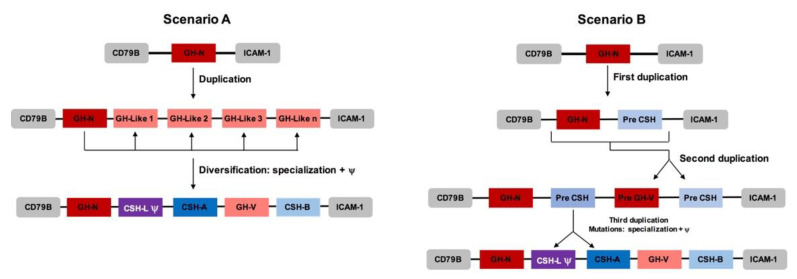
Two possible scenarios that gave rise to the modern *GH loci* in apes. The first scenario (**A**) considers one duplication event followed by mutations/specializations events. The second scenario (**B**) assumes two initial duplications and an additional round of duplication followed by mutations/specializations events.

**Table 1 genes-14-00241-t001:** Sequences used in this study.

Species	Accession Number
Human (*H. sapiens*)	J03071
Neanderthal (*H. sapiens neanderthalensis*)	GCA_000208225
Chimpanzee (*P. troglodytes*)	JN622009
Gorilla (*G. gorilla*)	KT971340
Orangutan (*P. abelii*)	KT959234
Gibbon (*N. leucogenys*)	CM016963.1
Wild board (*S. scrofa*)	GCF_000003025

**Table 2 genes-14-00241-t002:** Repetitive elements found in the human, Neanderthal, chimpanzee, gorilla, and orangutan *GH loci*.

	Human	Neanderthal	Chimpanzee	Gorilla	Orangutan
*Locus* length (bp)	75,316	75,287	84,237	98,181	63,107
*Alu* sense	22	28	15	29	18
*Alu* antisense	26	28	15	23	18
Retrotransposon	0	0	2	0	1

## Data Availability

Not applicable.

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
