# Peer review of "Gene Content and Coding Diversity of the Growth Hormone Loci of Apes"

_genes, 2023, doi:10.3390/genes14020241_

Round 1
Reviewer 1 Report
The paper addresses the genetic evolution of the GH locus in primates.
As a clinician it is not always easy to read this paper, because it is somewhat out of my comfort zone. Still there is also an issue with the English language or the use of jargon.
I will give some examples:
line 16 unclear sentence: please rephrase
line 38 "taxa" please explain or rephrase
Line 46 Cercopithecidae please rename in brackets like zou do now in line 54 afterwards
line 78 " said" primates please rephrase
line 218 "clade" do you mean "cluster" ?
More general remarks:
Paragraph lines 53-64: it would help to show a figure of a phylogenetic tree in stead of the text or in addition to the text.
Figure 4 is only partially visible and it would help the reader i fit is shown earlier in the discussion section.
Author Response
November 23, 2022
Prof. Dr. J. Peter W. Young
Editor-in-Chief of Genes
Department of Biology, University of York
Heslington, York YO10 5DD, UK
Dear Dr. Young:
We would like to render the improved version of the following:
Manuscript number: 2027451
Title: Gene content and coding diversity of the growth hormone loci of apes.
The manuscript has been improved and adjusted according to the recommendations provided.
Reviewer #1
The paper addresses the genetic evolution of the GH locus in primates. As a clinician, it is not always easy to read this paper, because it is somewhat out of my comfort zone. Still, there is also an issue with the English language or the use of jargon. I will give some examples:
Reviewer’s comment 1:
Line 16 unclear sentence: please rephrase.
Author’s answer 1:
The original sentence said: “it has remained unclear how the multigene family was favored feature”. It was rewritten as follows: “it has remained unclear how the multigene family was favored”.
Reviewer’s comment 2:
Line 38 "taxa" please explain or rephrase.
Author’s answer 2:
The original sentence said: “Even though there is considerable variation among members of this family in evolutionary rate across taxa and that distant members share only about 25 - 30% amino acid sequence identity” It was rewritten as follows: “Even though there is a considerable variation among members of this family in the evolutionary rate across human and non-human primates and their distant members share only about 25 - 30% amino acid sequence identity”.
Reviewer’s comment 3:
Line 46 Cercopithecidae please rename in brackets like you do now in line 54 afterwards.
Author’s answer 3:
The sentence in brackets “(Old-World monkeys and apes, including humans)” was replaced from line 54 to line 46.
Reviewer’s comment 4:
Line 78 “said” primates please rephrase
Author’s answer 4:
The original sentence said: “These features have been advanced as the basis for increased social complexity and cognitive capacity in said primates.” It was rewritten as follows: “These features have been advanced as the basis for increased social complexity and cognitive capacity in the primates.”
Reviewer’s comment 5:
Line 218 “clade” do you mean “cluster”?
Author’s answer 5:
Yes, we said “clade”. In phylogenetic terms, the word clade is used.
Reviewer’s comment 6:
More general remarks: Paragraph lines 53-64: it would help to show a figure of a phylogenetic tree instead of the text or in addition to the text.
Author’s answer 6:
An overview of gene composition and organization of GH loci of primates was added with the figure legend: “Fig. 1 Dramatic evolution of the GH locus in primates. At the bottom is the lemur’s (a prosimian) single gene locus (GH, blue box). Then into New World monkeys (illustrated by the spider monkey and the marmoset), Old World monkeys (exemplified by the baboon and the macaque), and apes (here demonstrated with the gorilla, the chimpanzee, and the human) the locus dramatically expands with the emergence of pseudogenes (Y), GH-V, and CSH genes (a compilation of previous results published by our group)”. All figures were renumbering.
Reviewer’s comment 7:
Figure 4 is only partially visible and it would help the reader if it is shown earlier in the discussion section.
Author’s answer 7:
Figure 4; now renumbering as figure 5, was adjusted to be visible.

Reviewer 2 Report
Dear Authors,
The paper is interested but it needs minor modification.
1. The methodology is not clear, e.g no of samples sequenced, no of sequence retrieved. Which type of sequencing plat form from NGS used is not clear. Data analysis methods, is not clearly descriped. E. g Phylogenetic analysis. I kept my comment at the document.
2. The result is not interpreted well especially, Phylogenetic part
3. No conclusion.
It will be nice if you incorporate the comment indicated in the main document.

Author Response
November 23, 2022
Prof. Dr. J. Peter W. Young
Editor-in-Chief of Genes
Department of Biology, University of York
Heslington, York YO10 5DD, UK
Dear Dr. Young:
We would like to return the improved and carefully revised version of the following manuscript:
Manuscript number: 2027451
Title: Gene content and coding diversity of the growth hormone loci of apes.
All parts of the manuscript were corrected following the reviewers´ recommendations. Please find in detail our response according to the suggestions pointed out by the second reviewer.
Reviewer #2
Dear Authors. The paper is interesting but it needs minor modification.
Reviewer’s comment 1:
The methodology is not clear, e.g. no. of samples sequenced, no. of sequences retrieved. Which type of sequencing platform from NGS was used is not clear. Data analysis methods are not clearly described. E. g Phylogenetic analysis. I kept my comment at the document.
Author’s answer 1:
Regarding sequenced samples, we used a single bacterial artificial chromosome (BAC) from each ape (chimpanzee, gorilla, and orangutan). These BACs were subject to New Generation Sequencing (NGS) using the Roche 454 sequencing platform. This methodology breaks the DNA into short fragments from 400 to 600 base pairs (bp). Then, adapters are linked to the fragments at their 5’ and 3’ ends. Followed by PCR amplification and finally sequencing. In this sense, in methodology, we changed the sentence: “The BAC’s GH-like gene sequences were amplified and cloned as previously described [25, 26] while gene, intergenic, and flanking sequences were obtained by NGS with the technical assistance of Genome Quebec.” By the sentence: “The GH-like gene BACs were sequenced by NGS using the Roche 454 platform as previously described [25, 26].”
Regarding the number of sequences retrieved, we download one sequence from each species. The retrieved sequences are indicated in table 1.
Regarding the NGS platform, we added the information that specifies it was the Roche 454 platform.
Regarding data analysis, specifically phylogenetic ones, the workflow for this analysis is as follows: First, in a text file the sequences (cDNA, DNA, or protein) are saved, then these files are uploaded in the Clustal Omega program, and with one click a text file with the alignment is done. Then, the file alignment is uploaded into the MEGA 6.06 software, one click in the construct phylogenetic tree option, and a phylogenetic tree is built. Neighbor-Joining (NJ) method is just one of many methods to build trees; we chose it because it is more suitable for the primate taxonomy. Bootstrap is a test that builds randomly 10, 100, or 1,000 trees to acquire the most accurate tree.
Reviewer’s comment 2:
The result is not interpreted well especially, Phylogenetic part.
Author’s answer 2:
We interpreted this analysis as follow: The phylogenetic analysis allowed us to postulate the hypothesis that gave rise the modern locus GH described in the discussion (Figure 5). This hypothesis is a new finding and we consider is mayor contribution of this work.
Reviewer’s comment 3:
No conclusion.
Author’s answer 3:
We added the following conclusion: “In this work, we determined the gene composition, organization, and proteins encoded in the GH loci of apes. The consensus arrangement of their genes is GH-N, CSH-A (variable number), GH-V, and CSH-B. The genes and proteins content is always a one-to-one proportion for the GH-N and GH-V type genes, while three-to-two, or two-to-one for the CSH type genes. Finally, we proposed two hypotheses to explain the origin of these multigenic loci. First, a duplication event followed by events of mutations/specializations. Second is an initial duplication, followed by mutations/specializations and an additional duplication event”.
Reviewer’s comment 4:
It will be nice if you incorporate the comment indicated in the main document.
Author’s answer 4:
All your comments were added to the manuscript.
